# Transient IL-33 upregulation in neonatal mouse lung promotes acute but not chronic type 2 immune responses induced by allergen later in life

Koji Iijima[1,2], Takao Kobayashi[1,2], Koji Matsumoto[1,2], Kenzo Ohara[1,2], Hirohito Kita[1,2], Li Y. Drake[1,3]*

1 Division of Allergic Diseases, Department of Medicine, Mayo Clinic, Rochester, Minnesota, United States of America, 2 Division of Allergy, Asthma, and Clinical Immunology, Department of Medicine, Mayo Clinic, Scottsdale, Arizona, United States of America, 3 Department of Anesthesiology and Perioperative Medicine, Mayo Clinic, Rochester, Minnesota, United States of America

* drake.li@mayo.edu

**Data Availability Statement:** All relevant data are within the manuscript and its Supporting Information files.

## Abstract

Early life respiratory insults, such as viral infections or hyperoxia, often increase asthma susceptibility later in life. The mechanisms underlying this increased susceptibility are not fully understood. IL-33 has been shown to be critically involved in allergic airway diseases. IL-33 expression in the neonatal lung can be increased by various respiratory insults associated with asthma development. Therefore, we investigated whether and how early life increases in IL-33 impact allergic airway responses later in life. Using a novel IL-33 transgenic mouse model, in which full-length IL-33 was inducible overexpressed in lung epithelial cells, we transiently upregulated lung IL-33 expression in neonatal mice for one week. After resting for 4–6 weeks, mice were intranasally exposed to a single-dose of recombinant IL-33 or the airborne allergen *Alternaria*. Alternatively, mice were exposed to *Alternaria* and ovalbumin multiple times for one month. We found that a transient increase in IL-33 expression during the neonatal period promoted IL-5 and IL-13 production when mice were later exposed to a single-dose of IL-33 or *Alternaria* in adulthood. However, increased IL-33 expression during the neonatal period did not affect airway inflammation, type 2 cytokine production, lung mucus production, or antigen-specific antibody responses when adult mice were exposed to *Alternaria* and ovalbumin multiple times. These results suggest that transient increased IL-33 expression early in life may have differential effects on allergic airway responses in later life, preferentially affecting allergen-induced acute type 2 cytokine production.

## Introduction

Allergic airway diseases often start in childhood [1]. Early life insults or distress in respiratory mucosa often increases the likelihood of developing asthma later in life. For example, early life

**Funding:** This work was supported by grant from the National Institute of Health NIH R01 HL117823. The funders had no role in study design, data collection and analysis, decision to publish, or preparation of the manuscript.

**Competing interests:** The authors have declared that no competing interests exist.

infection with viruses such as respiratory syncytial virus (RSV) and rhinovirus (RV), and exposure to hyperoxia due to oxygen supplementation therapy, are major risk factors for asthma development [2–4]. However, our knowledge is limited regarding the underlying mechanisms to explain how early life respiratory insults increase asthma susceptibility later in life.

IL-33, an IL-1 family cytokine, has been shown to be critically involved in allergic airway diseases [5]. IL-33 is constitutively expressed by a variety of cell types in the lung, including epithelial cells, fibroblasts, and endothelial cells [6, 7]. IL-33 is involved in many physiological and pathological processes [8]. The best known function of IL-33 is to activate cell types involved in type 2 immunity, including Th2 cells and group 2 innate lymphocytes (ILC2s) [9, 10]. As the innate counterpart of Th2 cells, ILC2s play critical roles in allergic airway diseases [5, 11]. At steady state, ILC2s are the primary target of IL-33 in the naïve mouse lung [12, 13]. In response to IL-33 stimulation, ILC2s rapidly produce large quantities of type 2 cytokines, such as IL-5, IL-13, and mediators involved in tissue repair, leading to eosinophilic airway inflammation and airway remodeling [11].

IL-33 expression in the lung starts as early as embryogenesis. Following postnatal lung inflation, IL-33 expression is upregulated and expression levels remain elevated during the first two weeks of life [14]. IL-33 expression levels in the neonatal lung can be further upregulated by various conditions such as environmental allergen exposure, hyperoxia due to oxygen supplementation therapy, and infection with respiratory viruses [15–18]. Limited information is available regarding whether and how such early-life increases in IL-33 impact lung immunity later in life.

We recently developed a novel conditional transgenic (tg) mouse model named CCSP-Il33tg mouse [19]. In this mouse, full-length IL-33 is transiently overexpressed in lung epithelial cells driven by the tetracycline-sensitive Club cell secretory protein (CCSP) promoter. We found that induced IL-33 overexpression in adult CCSP-Il33tg mice produced no pathologic lung effects at steady-state. In contrast, induced IL-33 overexpression in neonatal CCSP-Il33tg mice, up to postnatal day 14, increased mortality and lung pathology [19]. In the current study, we have used the CCSP-Il33tg mouse model to address the question of how increased IL-33 expression early in life affects allergic airway responses later in life. Our data show that transient IL-33 overexpression during the neonatal period enhanced acute type 2 cytokine responses after a single dose of allergen exposure later in life. However, increased IL-33 expression in neonatal lung did not affect allergic airway responses in adult mice repetitively exposed to allergens in a chronic course. Collectively these data suggest that IL-33 upregulation in the developing lung may preferentially influence acute but not chronic type 2 immune responses later in life.

## Materials and methods

### Mice

CCSP-Il33tg mice (FVB background) were generated at Mayo Clinic, Rochester, MN as described before [19]. In this mouse, expression of the gene encoding full-length IL-33 is induced by doxycycline (Dox)-sensitive rtTA protein driven by the rat CCSP promoter. To generate experimental mice, CCSP-Il33 double-tg mice were bred with CCSP-rtTA single-tg mice. Non-tg or single-tg littermates were used as controls for all experiments. To induce IL-33 transgene expression in pups, nursing mothers were provided with Dox (Mayo Pharmacy, Mayo Clinic, Rochester, MN) via drinking water (Dox was dissolved in 2.5% sucrose water and administered at 0.5 mg/ml) for the indicated periods. Dox-containing water was prepared fresh and changed twice weekly. Mice were observed daily, and moribund mice were euthanized immediately by intraperitoneal injection of pentobarbital. All animal experiments

and handling procedures were approved by the Mayo Clinic Institutional Animal Care and Use Committee and performed according to established guidelines.

## Reagents

Fluorescence-labeled antibodies to CD3 (2C11), CD4 (RM4-5), CD8 (53–6.7), CD11c (HL3), CD11b (M1/70), CD19 (1D3), CD49b (DX5), Gr1 (RB6-8C5), Ter119, CD25 (7D4), CD44 (IM7), CD16/CD32 (2.4G2), CD45R/B220 (RA3-6B2), CD23 (B3B4) IgG1, IgM, IgA, and IgE were purchased from BD Biosciences. The anti-T1/ST2 (DJ8) was from MD Biosciences (St. Paul, MN). Untagged recombinant IL-33 was from eBioscience.

## Airway allergen exposure models

Adult mice (6–8 week old) were lightly anesthetized with isoflurane prior to intranasal (i.n.) administration of *Alternaria* (*Alt*) extract (Greer Laboratories, Lenoir, NC) in 50 μl of endotoxin-free PBS. Control mice received 50 μl of PBS only. Two different allergen exposure models were used. For the acute allergen exposure model, mice were administered a single i.n. dose of *Alt* extract (50 μg/dose) under isoflurane anesthesia and euthanized either 1 hour later to collect bronchoalveolar lavage (BAL) fluid or 4.5 hours later to collect lungs. All mice were euthanized by intraperitoneal injection of pentobarbital. For the chronic allergen exposure model, mice were exposed i.n. to a mixture of *Alt* extract (20 μg/dose) and ovalbumin (OVA, 10 μg/dose) under isoflurane anesthesia twice a week for 4 weeks. Twenty-four hours after the last allergen exposure, mice were anesthetized by isoflurane and 100 μl blood was collected from each mouse by retroorbital bleed. After blood collection, mice were euthanized by intraperitoneal injection of pentobarbital, and BAL fluids as well as lung tissues were collected. The number of cells in BAL fluid was determined, and cell differentials were determined using cytospin slides stained with Wright-Giemsa. Lung tissues were homogenized in 1 ml of PBS. Cytokine levels in the lung homogenates were determined as described below. Some lung lobes were fixed in 10% formalin, embedded in paraffin, sectioned, and stained with hematoxylin and eosin (H&E) or Periodic acid–Schiff (PAS) reagents.

## Immunoassays

The levels of cytokines in lung homogenates and *in vitro* culture supernatants were analyzed using Quantikine or Duoset ELISA kits (R&D Systems), following the manufacturer's instructions. The levels of OVA-specific IgE or IgG1 in the plasma were measured via sandwich ELISA, as previously described [20].

## Cell culture

Mice were euthanized by intraperitoneal injection of pentobarbital and then mouse lungs were harvested. Mouse lungs were minced and digested with a cocktail of collagenases (Roche Diagnostics) and DNase I (StemCell Technologies) to obtain a single cell suspension. Red blood cells were lysed with ammonium chloride/potassium lysing buffer. Lung cells were resuspended in RPMI 1640 medium supplemented with 50 μM 2-ME, 100 units/ml penicillin, 100 μg/ml streptomycin, and 10% fetal bovine serum and then cultured in a 96-well tissue culture plate at $3 \times 10^5$ cells/well with or without IL-33 (10 ng/ml). Alternatively, ILC2s were isolated from mouse lungs. Lung cells were first enriched for ILC2s by EasySep® mouse ILC2 enrichment kit per the manufacturer's instructions (StemCell Technologies). The ILC2-enriched lung cells were then stained with a fluorescence-labeled lineage+ (Lin+) antibody cocktail, including CD3, CD4, CD8, CD11c, CD11b, CD19, CD16/CD32, CD49b, B220, Gr1,

and Ter119, together with anti-CD25, and anti-CD44. ILC2s were isolated as the Lin⁻CD25⁺CD44^hi cell population by FACS (BD FACSAria®). Sorted ILC2s were resuspended in RPMI 1640 medium as above and then cultured in a 96-well tissue culture plate at $2 \times 10^3$ cells/well with or without IL-33 (10 ng/ml). Culture supernatants were collected after 4 days. The levels of IL-5 and IL-13 in the supernatants were determined by ELISA.

## Statistical analysis

Data are presented as mean ± SEM, as indicated in the figure legends. Statistical significance was assessed using multiple t tests in grouped analyses in GraphPad software; $p < 0.05$ was considered significant. Outliers were determined using Grubbs' test in GraphPad software.

## Results

### Increased IL-33 expression elevates IL-5 levels in the neonatal lung

To study the function of IL-33 in the lung, we developed a novel transgenic mouse model in which full-length mouse IL-33 is overexpressed specifically in lung epithelial cells, namely CCSP-Il33tg mouse [19]. In this mouse model, IL-33 overexpression in the lungs is induced when the mice are given Dox water. Dox administration induced a 3 to 5-fold increase in lung levels of IL-33 protein in 10 day-old CCSP-Il33tg mice as compared with non-transgenic control mice [19] (Fig 1A). IL-33 protein was detectable by ELISA in BAL fluids of 10 day-old CCSP-Il33tg mice but not in BAL of control mice (Fig 1B). Previously we showed that IL-5 protein levels in neonatal lungs were slightly increased in CCSP-Il33tg mice, whereas IL-13 protein levels were comparable in CCSP-Il33tg mice and control mice [19]. In this study, we further analyzed IL-5 and IL-13 protein levels in BAL fluids from Dox-treated neonatal mice. Correlating with the increased IL-5 expression in lung tissue, low levels of IL-5 protein were detected in BAL fluids from 10 day old CCSP-Il33tg mice but not in control mice (Fig 1C). In contrast, IL-13 was not consistently detected in BAL fluids from the same mice (Fig 1D). Overall, these results indicate that induced IL-33 upregulation in CCSP-Il33tg mice increases IL-5 production and secretion in neonatal lungs.

### Transient increased IL-33 expression primes neonatal mice to develop robust innate type 2 cytokine responses later in adulthood

IL-33 expression in CCSP-Il33tg mice is Dox-dependent and IL-33 protein levels in Dox-treated CCSP-Il33tg mice return to the baseline levels within two days of cessation of Dox administration [19]. Thus, we used this mouse model to address whether increased IL-33 expression in the neonatal period impacts type 2 immune responses later in life. To this end, we administered Dox to nursing mothers to induce IL-33 overexpression for 7 days when the pups were between 6 to 13 days old. These mice were then rested for 4–6 weeks with normal drinking water. We term these mice 'neonatal-IL-33-experienced mice'. At the end of the resting period, these neonatal-IL-33-experienced mice were given a single i.n. dose of recombinant IL-33 at 6- to 8-weeks of age and the lung IL-5 and IL-13 levels were analyzed at 4.5 hours (Fig 2A). In both male and female mice, we found that PBS exposure induced minimal amounts of IL-5 and IL-13 production in the lungs of both neonatal-IL-33-experienced CCSP-Il33tg mice and non-transgenic littermate mice (Fig 2B and 2C), indicating that increased IL-33 expression in the neonatal period did not result in elevation of baseline levels of IL-5 and IL-13 later in life. Exposure to recombinant IL-33 induced significant increases in lung levels of both IL-5 and IL-13 in CCSP-Il33tg and non-transgenic control mice as compared to those exposed to

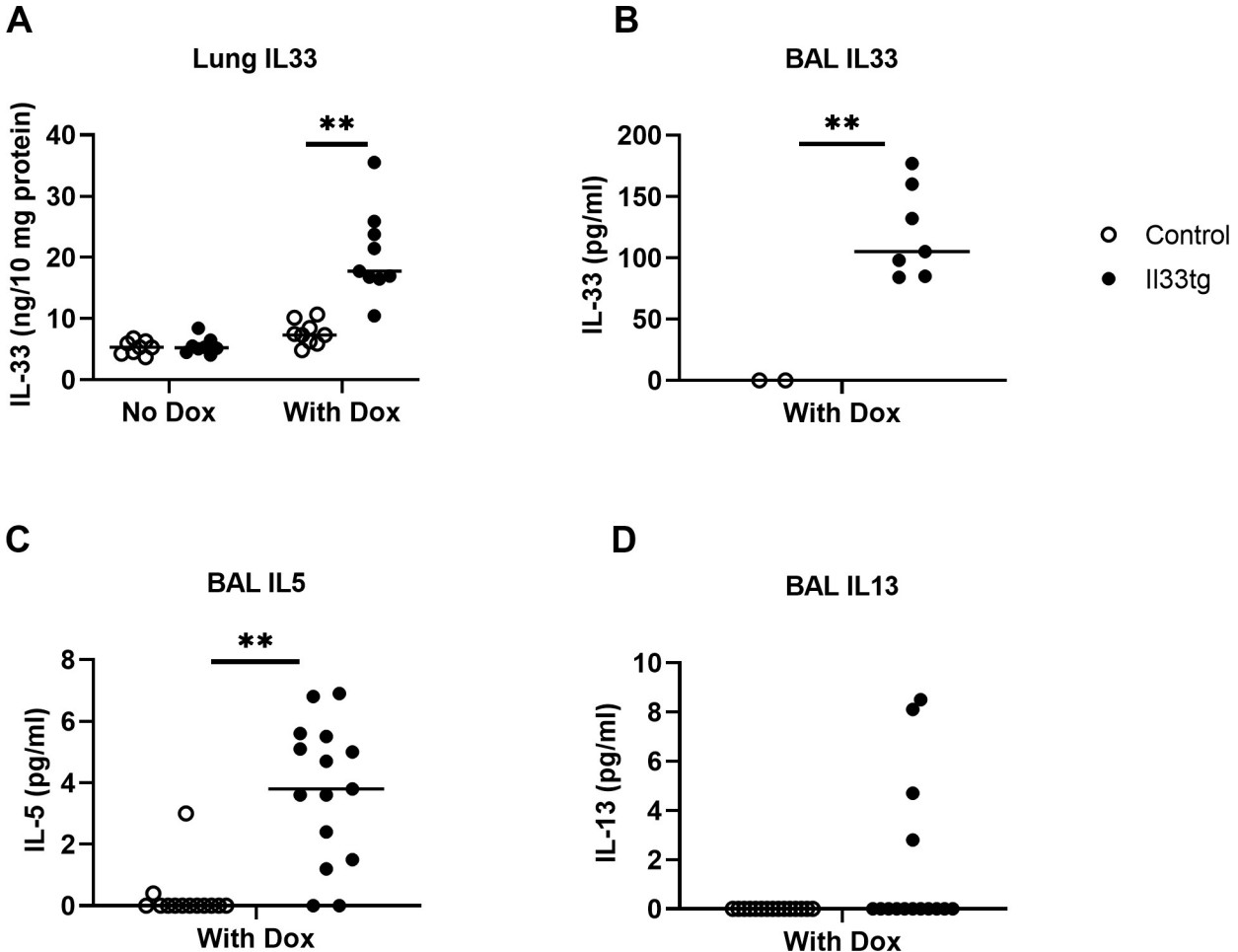

**Fig 1. Cytokine levels in neonatal mouse lungs.** CCSP-Il33tg mice or control mice were provided with Dox-containing water on postnatal days 1–10 and cytokines in lung tissue or BAL fluid were analyzed on day 10. (A, B) IL-33 levels in lung tissue or BAL fluid from mice treated with or without Dox. (C, D) IL-5 and IL-13 levels in BAL fluid from mice treated with Dox. Each dot represents one mouse (n = 7–16 mice per group, pooled from more than 2 independent experiments). **p < 0.01 between the groups, indicated by horizontal line.

PBS. Importantly, lung levels of IL-5 and IL-13 in CCSP-Il33tg mice were more than 2-fold higher than those in non-transgenic control mice (Fig 2B and 2C).

Extensive studies show that IL-33 plays a critical role in airway innate immune responses induced by allergens [12, 13, 21, 22]. To investigate whether increased IL-33 expression in the neonatal period affects allergen-induced innate type 2 immune responses in adulthood, we exposed neonatal-IL-33-experienced mice or control mice to a common airborne allergen *Altenaria (Alt)* and then analyzed type 2 cytokine responses by examining IL-5 and IL-13 levels at 4.5 hours post allergen exposure. Previous studies have shown that lung ILC2s are the primary cellular source of IL-5 and IL-13 production in this acute *Alt* exposure mouse model [12, 23], suggesting that the cytokine responses represent innate immune responses. Similar to IL-33-induced cytokine responses, *Alt*-induced IL-5 and IL-13 production in neonatal-IL-33-experienced CCSP-Il33tg mice were more than 2-fold higher than the response in non-transgenic control mice (Fig 2D and 2E). Collectively, these data suggest that transient increased IL-33 expression in the neonatal period enhances IL-33- or *Alt*-induced airway innate type 2 immune responses later in life.

## A

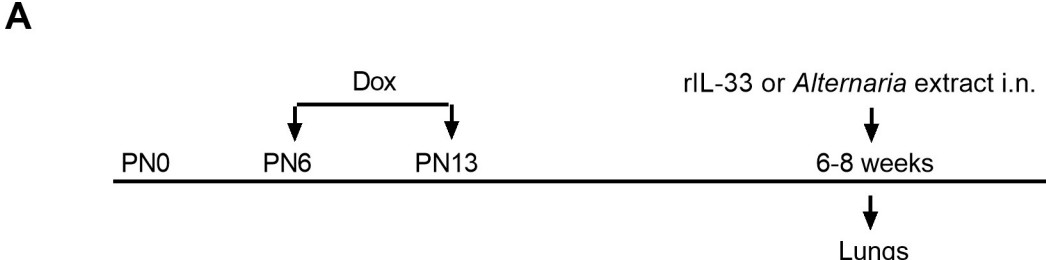

## B

**Lung IL5**

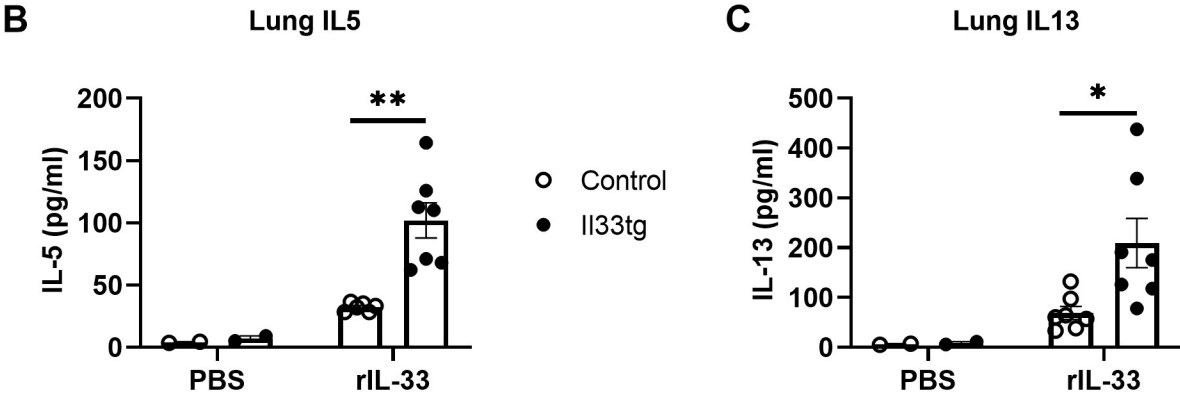

## C

**Lung IL13**

## D

**Lung IL5**

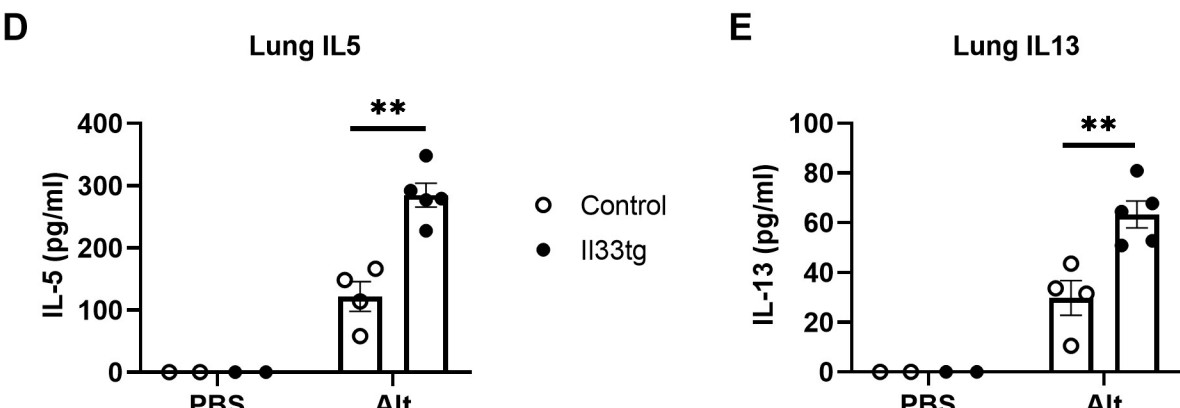

## E

**Lung IL13**

**Fig 2. Neonatal-IL-33-experienced mice had enhanced innate type 2 immune responses later in life.** (A) Mouse experimental protocol. Mice were provided with Dox water on postnatal days 6–13. After resting for 4–6 weeks, CCSP-Il33tg mice or control mice were i.n. exposed to either recombinant IL-33 (100 ng/mouse) or *Alt* extract (50 μg/mouse). Lung cytokines in these mice were analyzed after 4.5 hours. (B, C) Lung IL-5 and IL-13 levels after PBS or IL-33 exposure. (D, E) Lung IL-5 and IL-13 levels after PBS or *Alt* exposure. Data are presented as means $\pm$ SEMs (n = 2 mice per group for PBS exposure; n = 4–5 mice per group for IL-33 or *Alt* exposure). *p < 0.05, **p < 0.01 between the groups, indicated by horizontal line. Data are representative of two (*Alt* exposure) or three (IL-33 exposure) independent experiments.

## Increased IL-33 expression in the neonatal period does not affect *Alt*-induced production of IL-33 and other cytokines in adulthood

Next, we investigated the potential mechanisms underlying the more robust innate type 2 immune responses in neonatal-IL-33-experienced mice. Although lung epithelial cells are the

primary cell source for IL-33 production, these cells also express the IL-33 receptor (ST2) and therefore respond to IL-33 [24, 25]. Alternatively, epithelial cells can be affected by downstream products that are produced in response to IL-33, such IL-13. Furthermore, a recent study showed that epithelial cells can remember previous insults and generate more robust responses upon second challenge [26]. Thus, the possibility exists that increased IL-33 expression in neonatal lungs can prime lung epithelial cells to produce or release more IL-33 and other cytokines upon exposure to allergens later in life and subsequently induce more production of IL-5 and IL-13 by lung immune cells. To test this hypothesis, we exposed neonatal-IL-33-experienced CCSP-Il33tg mice or control mice to a single dose of *Alt* extract and subsequently collected BAL fluid at 1h (to measure IL-33 release) and lungs at 4.5 hours (to measure cytokine production), respectively. We found that neonatal-IL-33-experienced CCSP-Il33tg mice and control mice had similar levels of IL-33 in BAL fluid and lungs either at baseline or after *Alt* exposure (Fig 3A and 3B). In addition to IL-33, several other epithelium-derived cytokines also play important roles in the allergic immune response, including IL-25, TSLP and IL-1 [27]. To examine whether IL-33 priming during the neonatal period increases expression of these cytokines in the lungs, we analyzed their levels in PBS- or *Alt*-exposed adult mice at 4.5 hours post exposure. We found that expression of IL-25, TSLP and IL-1 cytokines was comparable in neonatal-IL-33-experienced CCSP-Il33tg mice and non-transgenic control mice (Fig 3C–3F). Collectively, these results suggest that the more robust innate type 2 immune responses to *Alt* in neonatal-IL-33-experienced mice are unlikely to have resulted from increased production or extracellular release of epithelium-derived cytokines.

## Increased IL-33 expression in the neonatal period does not change ILC2 numbers but enhance ILC2 function

Based on our previous finding that lung ILC2s are the primary cell source of IL-5 and IL-13 in the acute *Alt* exposure mouse model [12, 23], we focused our study on lung ILC2s. Using adult mice, Martinez-Gonzalez et al. showed that recombinant IL-33 i.n. exposure in adulthood induced innate memory in lung ILC2s [28]. A recent study showed that endogenous homeostatic levels of IL-33 may "train" lung ILC2s in the neonatal period and impact ILC2 function in adulthood [29]. We hypothesized that increased levels of lung IL-33 prime neonatal lung ILC2s to produce higher levels of IL-5 and IL-13 in response to *Alt* exposure later in life. To test this hypothesis, we examined the number and function of lung ILC2s in neonatal-IL-33-experienced mice. First, we quantified lung ILC2s in adult mouse lungs by flow cytometry. ILC2 numbers were similar in neonatal-IL-33-experienced CCSP-Il33tg mice and control mice at steady state (Fig 4A). Moreover, surface expression levels of IL-33 receptor (ST2) on ILC2s were also similar between the two groups (Fig 4B).

To study ILC2 function, we took an *in vitro* culture approach. We FACS-sorted lung ILC2s and then stimulated them with IL-33 *in vitro*. Sorted ILC2s from neonatal-IL-33-experienced CCSP-Il33tg mice produced significantly higher levels of IL-5 (approximately 2.2-fold) compared to ILC2s from control mice (Fig 4C). IL-33-experienced ILC2s also produced higher levels of IL-13 (approximately 1.4-fold) compared to control ILC2s (Fig 4D), but the differences were not statistically significant. Together these data suggest that transiently increased IL-33 expression in the neonatal period may "train" ILC2s to have a memory type response to IL-33 later in life.

## Increased neonatal IL-33 expression does not affect allergic immune responses induced by chronic allergen exposure later in life

To investigate how increased neonatal IL-33 expression impacts airway immune responses and inflammation induced by chronic allergen exposure, we exposed neonatal-

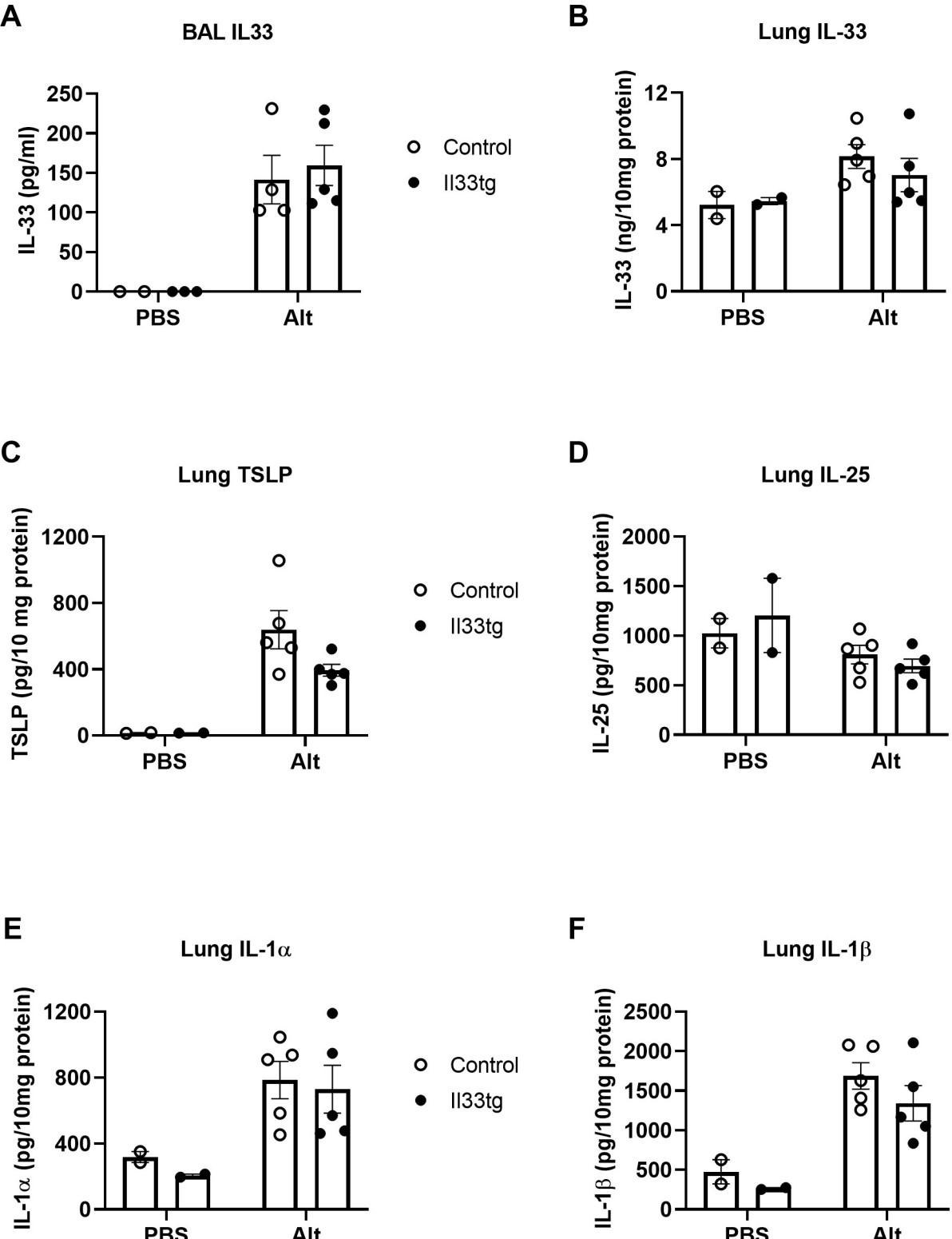

**Fig 3. *Alt* extract induced comparable levels of IL-33 airway lumen release and lung epithelial cytokine production in neonatal-IL-33-experienced mice and control mice.** Mice were provided with Dox water on postnatal days 6–13. After resting for 4–6 weeks, the mice were i.n. exposed to *Alt* extract (50 μg/mouse). BAL IL-33 (A) was analyzed at 1 hour post *Alt* exposure. Lung cytokines (B-F) were analyzed at 4.5 hours post *Alt* exposure. Data are presented as means ± SEMs (n = 2 mice per group for PBS exposure; n = 4–5 mice per group for *Alt* exposure). Data are representative of two independent experiments.

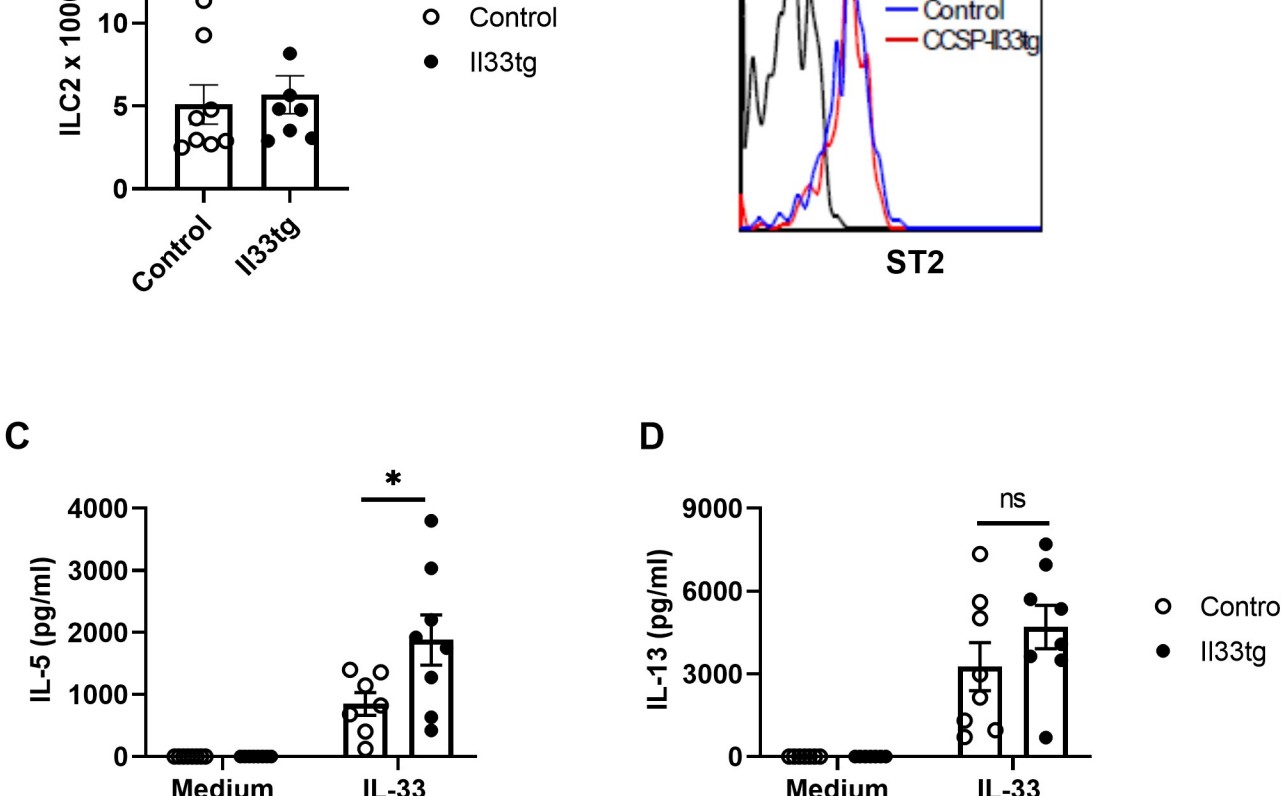

**Fig 4. ILC2 function but not number was enhanced in neonatal-IL-33-experienced mice.** Mice were provided with Dox water on postnatal days 6–13. After resting for 4–6 weeks, lung ILC2s were analyzed. (A) The number of lung ILC2s in neonatal-IL-33-experienced CCSP-Il33tg mice or control mice were quantified by flow cytometry (n = 8 mice per group). (B) ST2 expression by lung ILC2s was analyzed by flow cytometry. (C) (D) FACS sorted ILC2s were cultured with IL-33 for 4 days. IL-5 and IL-13 levels in culture supernatants were determined. Data are pooled from 3 independent experiments. Each dot represents one well in 96-well plate.

IL-33-experienced CCSP-Il33tg mice and control mice to a combination of *Alt* extract and OVA (*Alt*/OVA) twice weekly for 4 weeks. Plasma, BAL and lung specimens were collected 24 h after the last allergen exposure (Fig 5A). First, we examined IL-33 protein levels in the lungs. Multiple exposures to *Alt*/OVA induced an approximately 3-fold increase in lung IL-33 levels compared to PBS-exposed mice. However, lung IL-33 protein levels were not significantly different between neonatal-IL-33-experienced CCSP-Il33tg mice and control mice (Fig 5B), indicating that increased IL-33 expression in the neonatal period did not affect allergen-induced IL-33 production later in life. Next, we examined the lung inflammation. Multiple exposures to *Alt*/OVA induced a robust increase in BAL eosinophils at 4 weeks, with eosinophils comprising approximately 90% of total BAL cells (Fig 5C). However, there were no significant differences in the level of BAL eosinophils between neonatal-IL-33-experienced CCSP-Il33tg mice and control mice (Fig 5C). Moreover, H&E and PAS staining revealed that *Alt*/OVA multiple exposures induced similar levels of leukocyte infiltration and PAS-positive airway epithelial cells in neonatal-IL-33-experienced CCSP-Il33tg mice and control mice (Fig 5D).

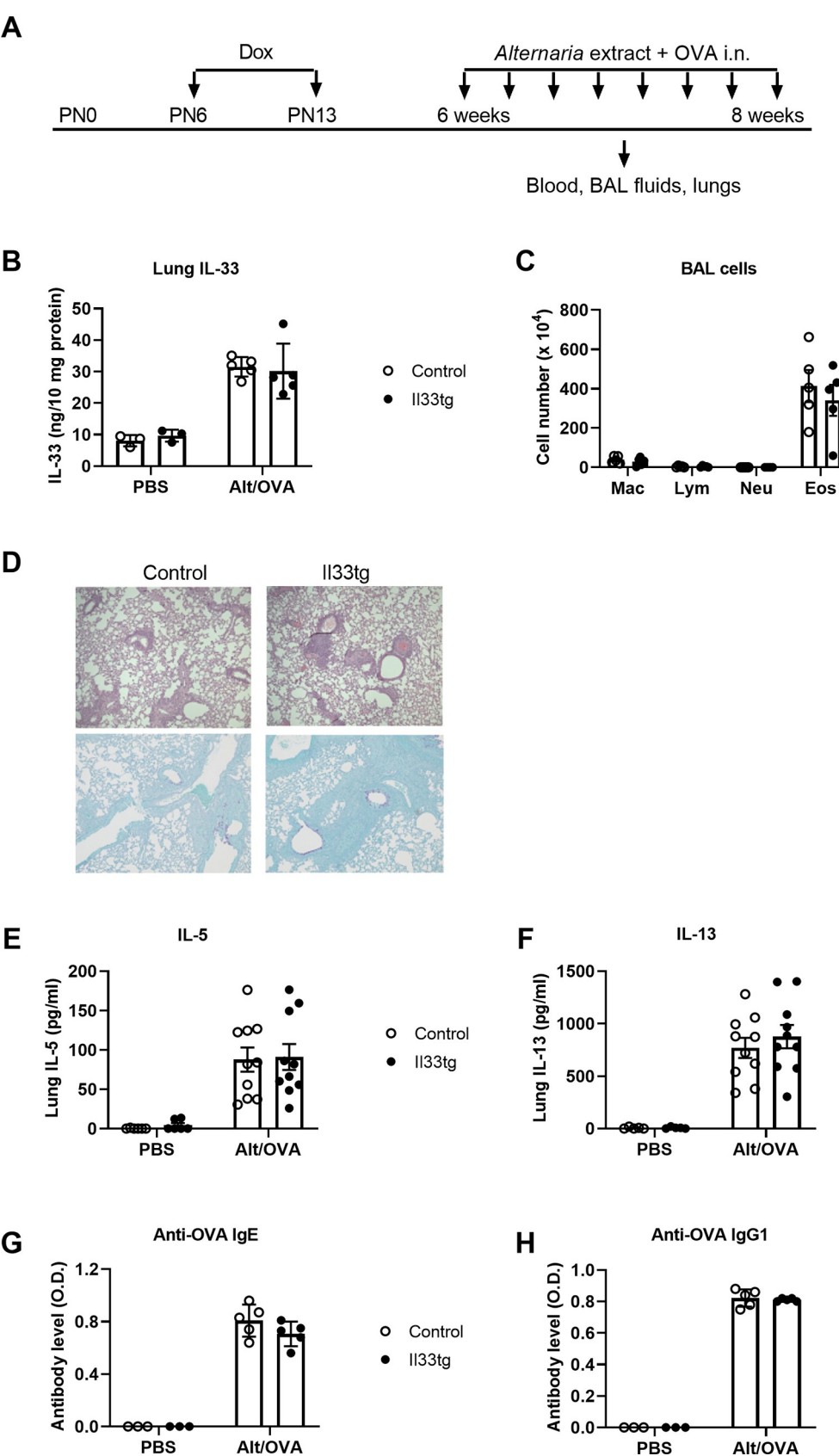

**Fig 5. Chronic exposure to *Alt*/OVA induced comparable airway immune responses in neonatal-IL-33-experienced mice and control mice.** (A) Mouse experimental protocol. Mice were provided with Dox water on postnatal days 6–13. After resting for 4–6 weeks, CCSP-Il33tg mice or control mice were i.n. exposed to PBS or a mixture of *Alternaria* and OVA twice a week for 4 weeks. Specimens were collected 24 hours after the last exposure. (B) IL-33 levels in lung homogenate were determined by ELISA. (C) Cells in BAL fluid were counted and presented as means ± SEM; n = 5 mice per group. Mac: macrophages; Lym: lymphocytes; Neu: neutrophils; Eos: eosinophils. (D) Lung sections were stained with H&E (top panels) or PAS (bottom panels). Original magnification, 160X. Data are representative of five mice per group. (E, F) Cytokine levels in lung homogenate were determined by ELISA. Data are pooled from two independent experiments with five mice per group. (G, H) Antibody titers in plasma were quantified by ELISA. Data are representative of two independent experiments with five mice per group.

Although a single dose of IL-33 or *Alt* exposure induced higher levels of IL-5 and IL-13 production in neonatal-IL-33-experienced CCSP-Il33tg mice (Fig 2), multiple exposures to *Alt*/OVA induced similar levels of lung IL-5 and IL-13 in neonatal-IL-33-experienced CCSP-Il33tg mice and control mice (Fig 5E and 5F). It has been shown previously that both ILC2s and CD4[+] T cells are the cellular sources of allergen-induced IL-5 and IL-13 production and that CD4[+] T cells likely contribute to cytokine production more than ILC2s in this chronic allergen exposure model [30]. Our results suggest that transiently increased IL-33 expression in neonatal lung may have little impact on adaptive immune cells.

To specifically analyze the adaptive immune responses after *Alt*/OVA multiple exposures, we examined antigen-specific antibody responses. *Alt*/OVA multiple exposures induced similar levels of OVA-specific IgE and IgG1 antibodies in neonatal-IL-33-experienced CCSP-Il33tg mice and control mice (Fig 5G and 5H). Collectively, these data suggest that increased IL-33 expression in neonatal lung does not affect chronic allergen exposure-induced airway immune responses, inflammation and pathology in later life.

## Discussion

Early life airway exposure to various respiratory insults, such as allergens, hyperoxia and viruses, is associated with increased risk of asthma development in later life [2–4]. These respiratory insults often induce increased IL-33 expression in neonatal lungs [15–18]. The current study suggests that increased IL-33 lung expression early in life likely promotes allergen-induced innate immune responses later in life. Using an IL-33 transgenic mouse model, lung IL-33 overexpression was transiently induced by doxycycline administration for one week during the neonatal period. When exposed to a single dose of recombinant IL-33 or allergen *Alternaria* later in adulthood, neonatal-IL-33-experienced mice showed enhanced IL-5 and IL-13 production in the airway 4.5 hours post exposure. Previous studies have shown that IL-33-responsive ILC2s are the primary cell source of IL-5 and IL-13 in the acute innate phase of *Alternaria*-induced cytokine responses [12, 23]. Thus, the more robust acute type 2 cytokine production observed in neonatal-IL-33-experienced mice likely reflects enhanced ILC2 function. Indeed, isolated lung ILC2s from neonatal-IL-33-experienced mice produced more IL-5 and IL-13 than ILC2s from control mice when stimulated with IL-33 *in vitro* (Fig 4). These data suggest that increased IL-33 expression in the neonatal lung primes ILC2s to develop more robust innate type 2 cytokine responses to acute IL-33 or *Alternaria* exposure later in life.

Accumulating evidence suggests that similar to the adaptive immunity, innate immune cells such as monocytes, macrophages, and natural killer cells, can also build immunological memory to mount increased responses to secondary stimulation [31]. Recent studies suggest that ILC2s can acquire immune memory as well. In an adult mouse model, IL-33- or allergen-experienced lung ILC2s produced more robust type 2 immune responses upon secondary challenges [28]. Using IL-33-deficient mice, it has been shown that endogenous homeostatic

levels of IL-33 in the neonatal lung may "train" ILC2s to respond more efficiently to IL-33 later in life [29]. The current study has further shown that increased levels of IL-33 in neonatal lung primed lung ILC2s to produce more potent allergen-induced innate type 2 immune responses in adulthood when the adult mice are exposed to a single dose of allergen. The more robust innate type 2 cytokine production in our neonatal-IL-33-experienced mice was likely due to intrinsic functional changes in ILC2s, as ILC2 numbers in these mice were not changed and FACS sorted ILC2s had higher responses to IL-33 stimulation *in vitro* (Fig 4). Recent studies demonstrate that epigenetic modification of inflammation-associated genes plays a major role in the innate immune memory in monocytes, macrophages, and natural killer cells [31]. Future studies are required to determine whether a similar mechanism applies to neonatal-IL-33-experienced lung ILC2s.

While one-time *Alternaria* exposure-induced acute type 2 cytokine production was increased in neonatal-IL-33-experienced mice, multiple *Alt*//OVA exposure provided over a one-month period did not result in significant differences in airway immune responses between neonatal-IL-33-experienced mice and control mice. Using a similar chronic allergen exposure mouse model, it has been shown previously that both ILC2s and CD4+ T cells contribute to allergen-induced IL-5 and IL-13 production and that no other cell types appear to express these cytokines in allergen-exposed mice [30]. When cell numbers were compared, IL-5 and IL-13 expressing-CD4+ T cells were approximately five-fold higher than IL-5 and IL-13 expressing-ILC2s [30]. Based on these previous findings, we speculate that IL-5 and IL-13 were mostly produced by CD4+ T cells in the chronic allergen exposure model used in this study. Since IL-5 and IL-13 levels were similar between control and neonatal-IL-33-experienced mice after chronic allergen exposure (Fig 5), the function of adult CD4+ T cells appears to have been unaffected by increased IL-33 expression in the neonatal period in our mouse model. Moreover, *Alt*/OVA-induced antibody responses were comparable between control and neonatal-IL-33-experienced mice (Fig 5), suggesting that increased IL-33 expression in the neonatal lung did not have significant effects on B cell function in adulthood in our mouse model.

Studies have shown that ILC2s interact with adaptive immune cells and promote T cell and B cell functions [13, 32–34]. In an adult mouse model, IL-33-experienced ILC2s were shown to be more efficient in driving Th2 cell differentiation [28]. In our study, although neonatal-IL-33-experienced ILC2s produced more type 2 cytokines, these ILC2s did not promote stronger type 2 immune responses after chronic allergen exposure in adulthood. There are many differences between our model and the adult mouse model, including allergen nature (*Alternaria* versus papain), the time window for IL-33 exposure (continuously for 7 days during the neonatal period versus once daily for 3 days in adulthood), endogenously overexpressed IL-33 versus recombinant IL-33, the amounts and characteristics of IL-33 (i.e. full-length vs processed), and allergen exposure models (3 times per week for 4 weeks versus one or two times only). Another notable difference is that one month after IL-33 exposure, ILC2 numbers were higher in IL-33-experienced mice in the adult mouse model but not in our IL33tg mouse model [28] (Fig 4A). It is possible that increased IL-33 expression in neonatal IL33tg mice only primed ILC2s to generate weak memory responses later in life; in turn, this weak memory response was overridden by a robust adaptive immune response when mice were exposed to allergen over a one month period.

In summary, the current study indicates that increased lung IL-33 expression during early development may enhance allergen-induced acute type 2 cytokine responses later in life. Early life exposure to allergens, oxygen supplementation and viral infections may increase the risk for asthma development later in life through IL-33-induced innate memory. Our data, together with that from a previous adult mouse study [28], suggest that IL-33-induced innate memory

affects allergic immune responses in a context-dependent manner that the nature and dose of allergens and frequency of allergen exposure may all play a role in the immune response. Future studies are required to understand the regulation of this complex role for IL-33 and the impact of IL-33-induced innate memory in human asthma. Collectively, our data suggest that increased IL-33 expression in the neonatal period preferentially influences type 2 immune responses in the acute innate phase after a single exposure to allergen, rather than type 2 immune responses in the chronic phase after multiple exposures to allergens.

## Author Contributions

**Conceptualization:** Hirohito Kita, Li Y. Drake.

**Data curation:** Koji Iijima, Takao Kobayashi, Kenzo Ohara, Li Y. Drake.

**Formal analysis:** Hirohito Kita, Li Y. Drake.

**Funding acquisition:** Hirohito Kita.

**Investigation:** Koji Iijima, Takao Kobayashi, Koji Matsumoto, Kenzo Ohara, Li Y. Drake.

**Methodology:** Koji Iijima, Takao Kobayashi, Koji Matsumoto, Kenzo Ohara, Li Y. Drake.

**Project administration:** Hirohito Kita, Li Y. Drake.

**Resources:** Hirohito Kita.

**Supervision:** Hirohito Kita, Li Y. Drake.

**Validation:** Li Y. Drake.

**Writing – original draft:** Li Y. Drake.

**Writing – review & editing:** Koji Iijima, Takao Kobayashi, Koji Matsumoto, Kenzo Ohara, Hirohito Kita, Li Y. Drake.

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
