## [Decision Letter · Decision Letter 0]

7 Jan 2021

PONE-D-20-38939

IL-33 upregulation in neonatal mouse lung promotes allergen-induced innate type 2 cytokine responses in later life

PLOS ONE

Dear Dr. drake,

Thank you for submitting your manuscript to PLOS ONE. After careful consideration, we feel that it has merit but does not fully meet PLOS ONE’s publication criteria as it currently stands. Therefore, we invite you to submit a revised version of the manuscript that addresses the points raised during the review process.

1. Clarify what new data are shown in current manuscript as compared to what was published before (reference 19).

2. Make sure the statements in the Abstract and Results sections correspond the actual data shown in Figures.

3. Correct all questionable statements throughout the manuscript's text.

4. Perform additional experiments suggested by the reviewer (24h and 48h timepoints).

5. Address the reviewer's comments about the number of animals tested for data shown in Figure 4.

We look forward to receiving your revised manuscript.

Kind regards,

Svetlana P. Chapoval

Academic Editor

PLOS ONE

Journal Requirements:

Additional Editor Comments :

1. Editor admits poor quality of all figures.

2. Be more specific about the number of experiments performed.

3. TSLP levels after Alternaria exposure shown in Figure 3 look statistically different between control and CCSP-IL33tg groups.

4. Specify if recombinant IL-33 was tagged.

5. Clarify what software was used for statistical analysis and how the statistical differences between experimental groups were calculated.

Journal Requirements:

2. We note that your study design may include death of a regulated animal as a likely outcome or planned experimental endpoint. At this time, we request that you please report additional details in your Methods section regarding animal care, as per our editorial guidelines (http://journals.plos.org/plosone/s/submission-guidelines#loc-humane-endpoints).      

For easy reference, we have attached a checklist that may be relevant for your submission. Please complete all items on the checklist at the following link:   http://journals.plos.org/plosone/s/file?id=bb1d/plos-one-humane-endpoints-checklist.docx         

Please upload the completed checklist as file type “Other” when resubmitting your manuscript. This document is for internal journal use only and will not be published if your article is accepted. We very much appreciate your attention to these requests and support of improved reporting standards in PLOS ONE submissions.

Reviewers' comments:

Reviewer's Responses to Questions

**Comments to the Author**

1. Is the manuscript technically sound, and do the data support the conclusions?

Reviewer #1: Partly

2. Has the statistical analysis been performed appropriately and rigorously? 

Reviewer #1: No

3. Have the authors made all data underlying the findings in their manuscript fully available?

Reviewer #1: Yes

4. Is the manuscript presented in an intelligible fashion and written in standard English?

Reviewer #1: Yes

5. Review Comments to the Author

Reviewer #1: This report is based on interesting data obtained in an interesting model, but additional experiments need to be performed and, most importantly, the interpretation of the data as well as the conclusions need to be revisited.

1. This manuscript substantially overlaps with the already published report from this group PMID 31471525 (ref. 19). The mouse model as well as the style of and type of data in the figures appear similar. The authors need to modify the manuscript to strongly and clearly emphasize the conceptual differences from the published report, both in the Introduction and the Discussion. It appears that these differences between the published report and this manuscript relate both to the timing of IL-33 induction as well as to the meaning of the findings? The reader should not be forced to read the previous publication in order to decipher the differences between the earlier paper and the current manuscript. This new paper should be able to stand on its own feet, requiring the suggested additional narration.

2. Furthering the concern above, the findings in the current report do not appear to be novel. This team’s previous work in ref. 19 suggested that the “increased expression of IL-33 in lung epithelial cells during the neonatal period causes lung pathology and that the mice are more vulnerable during the first 2 wk of life.” The findings in the current manuscript appear very similar. The distinct novelty of the current report needs to be explicitly stated in the Abstract, Introduction, and Discussion, by contrasting with ref. 19, e.g., “We now additionally report that …”

3. The conclusions of the work do not appear to be justified, and the Abstract is outright misleading in the major claim that “increased IL-33 expression during the neonatal period did not affect airway inflammation, type 2 cytokine production, lung mucus production, or antigen-specific antibody responses.” Although there is seemingly no effect shown in figure 5, these data reflect the “terminal,” “oversaturated” process driven by repeated, prolonged Alt / OVA exposure. By contrast, data in figure 2, which are arguably as pathophysiologically relevant as the data in figure 5, clearly shows that IL-5 and IL-13 were strongly upregulated in neonatal-IL-33-experienced mice. The data in figure 4 also indicate a likely effect (see below) on IL-5 and IL-13 production, which, if true, would implicate the contributions from ILC2 and offer a convincing mechanism for the observed phenomenology. Further misleading the reader, the statement “that innate type 2 immune responses may be affected more robustly than adaptive type 2 responses” is speculative. Additionally, the designation of “adaptive” vs “innate” responses in this context is unclear and misleading. The conclusions of the work should be built on the entirety of the data and not on an arbitrarily selected subset of the findings.

4. In figure 2, additional animals should have been tested 24 and 48 hours after rIL-33 and Alt extract exposures and analyzed for IL-33, IL-5, and IL-13 levels, differential BAL cell counts, and histological changes, similar to panels 5B – E. The findings would likely lead to different conclusions of the overall project.

5. There is a serious concern with data analyses and interpretation across the report. Figure 1 shows highly variable elevations in IL-5 in the majority of the animals, as well as variable, and in some animals substantial, elevation in IL-13. These data suggest that in some, but not all, animals the transgenic full-length IL-33 underwent spontaneous activation into mature IL-33 cytokine; it is well known that full-length IL-33 does not have a pro-Th2 effects whereas mature IL-33 does. What is the mechanism of such spontaneous and highly heterogenous IL-33 maturation? More importantly, with such strong heterogeneity of type 2 activation, the Dox-treated animals should not be viewed as a uniformed group. It is therefore concerning that in all subsequent figures, bar graphs are shown instead of scatterplots showing individual animals similar to figure 1. This is further a concern because in the important figure 4, central conclusions regarding “slightly higher values” and lack of statistical significance are drawn based on n = 3 per group. It is likely that with more animals tested, larger and significant differences would be observed, driving a different interpretation of the meaning of the findings.

6. PLOS authors have the option to publish the peer review history of their article (what does this mean?). If published, this will include your full peer review and any attached files.

Reviewer #1: No

---

## [Author Response · Author response to Decision Letter 0]

18 Mar 2021

Editor’s comments

1. Clarify what new data are shown in current manuscript as compared to what was published before (reference 19).

Response: The data in Fig 1A and Fig 1B were presented in a different format in reference 19. We think that it is necessary to show these data again in the current study to introduce the Il33tg mouse model. Other than these data, all additional presented data are novel unpublished findings. 

2. Make sure the statements in the Abstract and Results sections correspond the actual data shown in Figures.

Response: Based on the reviewer’s suggestions, we have modified some of the statements. The changes are highlighted in yellow.

3. Correct all questionable statements throughout the manuscript's text.

Response: Please see the response above.

4. Perform additional experiments suggested by the reviewer (24h and 48h timepoints).

Response: Our preliminary time course experiments show that cytokine production peaks at 3-6h after a single dose of allergen exposure and that very little cytokine can be detected at 24h (Fig 1 for reviewer). Therefore, we do not believe including data at 24h/48h is of value to the manuscript. 

5. Address the reviewer's comments about the number of animals tested for data shown in Figure 4.

Response: All the comments have been addressed as shown below.

Additional Editor Comments:

1. Editor admits poor quality of all figures.

Response: We have re-made all the figures using different software. We hope that the quality of these new figures meets the standards of Plos One.

2. Be more specific about the number of experiments performed.

Response: We have added more information in figure legends. 

3. TSLP levels after Alternaria exposure shown in Figure 3 look statistically different between control and CCSP-IL33tg groups.

Response: The t test using grouped analyses in GraphPad did not reveal a significance difference between the two groups.

4. Specify if recombinant IL-33 was tagged.

Response: It was not tagged. We have added this information in the manuscript (Line 102).

5. Clarify what software was used for statistical analysis and how the statistical differences between experimental groups were calculated.

 Response: We have updated the Methods section as follows: “Statistical significance was assessed using multiple t tests in grouped analyses in GraphPad software; p < 0.05 was considered significant. Outliers were determined using Grubbs' test in GraphPad software.”

Journal Requirements:

Response: We have revised the manuscript accordingly. 

2. We note that your study design may include death of a regulated animal as a likely outcome or planned experimental endpoint. At this time, we request that you please report additional details in your Methods section regarding animal care, as per our editorial guidelines (http://journals.plos.org/plosone/s/submission-guidelines#loc-humane-endpoints). 

Response: We have added more information in the Methods section following the editorial guidelines. Please see the highlighted sentences.

Reviewer’s comments

Reviewer #1: 

Comment 1. This manuscript substantially overlaps with the already published report from this group PMID 31471525 (ref. 19). The mouse model as well as the style of and type of data in the figures appear similar. The authors need to modify the manuscript to strongly and clearly emphasize the conceptual differences from the published report, both in the Introduction and the Discussion. It appears that these differences between the published report and this manuscript relate both to the timing of IL-33 induction as well as to the meaning of the findings? The reader should not be forced to read the previous publication in order to decipher the differences between the earlier paper and the current manuscript. This new paper should be able to stand on its own feet, requiring the suggested additional narration.

Response: The current manuscript has utilized the Il33tg mouse model published previously to address a new question, i.e., how transiently increased IL-33 expression early in life affects allergic airway responses later in life. This has been described in the introduction (Lines 72-74). We agree that not enough information was provided in our original manuscript submission regarding the previous publication. We have added more information to the revised manuscript (Lines 69-72). Although the data in Fig 1A and Fig 1B were presented in a different format in the previous report, we believe these data are necessary to introduce the Il33tg mouse model. Apart from these data, all presented data are novel unpublished findings. 

Comment 2. Furthering the concern above, the findings in the current report do not appear to be novel. This team’s previous work in ref. 19 suggested that the “increased expression of IL-33 in lung epithelial cells during the neonatal period causes lung pathology and that the mice are more vulnerable during the first 2 wk of life.” The findings in the current manuscript appear very similar. The distinct novelty of the current report needs to be explicitly stated in the Abstract, Introduction, and Discussion, by contrasting with ref. 19, e.g., “We now additionally report that …”

Response: The current report has utilized a completely different experimental approach than the previous report as indicated in the diagram in Fig. 2A. The previous report analyzed the neonatal mice immediately after the induction of IL-33 overexpression. In the current report, IL-33 overexpression was induced in neonatal mice which were then rested for 4-6 weeks as described in the Results section (Lines 185-190). As mentioned above, apart from the data in Fig 1A and Fig 1B, all additional data are novel unpublished findings. 

We would like to point out that our conclusion in the current manuscript is not that “increased expression of IL-33 in lung epithelial cells during the neonatal period causes lung pathology and that the mice are more vulnerable during the first 2 wk of life.” That was the conclusion in our previous publication. As we have described in the abstract, we have concluded in the current report that “transient increased IL-33 expression early in life may have differential effects on allergic airway responses in later life, preferentially affecting allergen-induced acute type 2 cytokine production.”

Comment 3. The conclusions of the work do not appear to be justified, and the Abstract is outright misleading in the major claim that “increased IL-33 expression during the neonatal period did not affect airway inflammation, type 2 cytokine production, lung mucus production, or antigen-specific antibody responses.” Although there is seemingly no effect shown in figure 5, these data reflect the “terminal,” “oversaturated” process driven by repeated, prolonged Alt / OVA exposure. By contrast, data in figure 2, which are arguably as pathophysiologically relevant as the data in figure 5, clearly shows that IL-5 and IL-13 were strongly upregulated in neonatal-IL-33-experienced mice. The data in figure 4 also indicate a likely effect (see below) on IL-5 and IL-13 production, which, if true, would implicate the contributions from ILC2 and offer a convincing mechanism for the observed phenomenology. Further misleading the reader, the statement “that innate type 2 immune responses may be affected more robustly than adaptive type 2 responses” is speculative. Additionally, the designation of “adaptive” vs “innate” responses in this context is unclear and misleading. The conclusions of the work should be built on the entirety of the data and not on an arbitrarily selected subset of the findings.

Response: This manuscript has two major conclusions. The first one is that “a transient increase in IL-33 expression during the neonatal period promoted IL-5 and IL-13 production when mice were later exposed to a single-dose of IL-33 or Alternaria in adulthood”. The second one is that “increased IL-33 expression during the neonatal period did not affect airway inflammation, type 2 cytokine production, lung mucus production, or antigen-specific antibody responses when adult mice were exposed to Alternaria and ovalbumin multiple times”. Both conclusions are described in the Abstract. The first conclusion is supported by Fig 2 and the second conclusion is supported by Fig 5. The data in Fig 4 has provided some insight into the mechanism underlying the first conclusion. 

We agree with the reviewer that the designation of “adaptive” vs “innate” responses in our experiments is unclear and perhaps inaccurate. To clarify our messages, we have added additional information regarding the two different experimental systems used in this study. Instead of innate and adaptive responses, we have changed the description to “acute type 2 responses” and “chronic type 2 responses”. 

Comment 4. In figure 2, additional animals should have been tested 24 and 48 hours after rIL-33 and Alt extract exposures and analyzed for IL-33, IL-5, and IL-13 levels, differential BAL cell counts, and histological changes, similar to panels 5B – E. The findings would likely lead to different conclusions of the overall project.

Response: Our preliminary time course experiments show that cytokine production peaks at 3-6h post allergen or IL-33 exposure and that very little cytokine can be detected at 24h (Fig 1 for reviewer). Therefore, we do not believe including data at 24h/48h is of value to the manuscript . 

5. There is a serious concern with data analyses and interpretation across the report. Figure 1 shows highly variable elevations in IL-5 in the majority of the animals, as well as variable, and in some animals substantial, elevation in IL-13. These data suggest that in some, but not all, animals the transgenic full-length IL-33 underwent spontaneous activation into mature IL-33 cytokine; it is well known that full-length IL-33 does not have a pro-Th2 effects whereas mature IL-33 does. What is the mechanism of such spontaneous and highly heterogenous IL-33 maturation? More importantly, with such strong heterogeneity of type 2 activation, the Dox-treated animals should not be viewed as a uniformed group. It is therefore concerning that in all subsequent figures, bar graphs are shown instead of scatterplots showing individual animals similar to figure 1. This is further a concern because in the important figure 4, central conclusions regarding “slightly higher values” and lack of statistical significance are drawn based on n = 3 per group. It is likely that with more animals tested, larger and significant differences would be observed, driving a different interpretation of the meaning of the findings.

Response: In our previous report, we found that induced IL-33 overexpression in neonatal mice resulted in variable lung pathology and mortality among the same littermate mice. We think these findings may be caused by variable milk intake among the pups within the same litter since the pups receive doxycycline through nursing. Despite such likely variability, we detected higher levels of IL-33 in BAL fluids and lungs from all Il33tg pups (Fig 1A, B) and these Il33tg mice showed a distinct phenotype when compared to control mice (ref 19). Therefore, we believe that it is reasonable to view the Il33tg mice as a uniform group. The IL-5 and IL-13 levels shown in Fig 1 are extremely low when compared to the IL-5 and IL-13 levels in Fig 2. The observed variability among pups in Fig 1 is caused in part by the fact that these low cytokine levels are very close to the detection limits of the ELISA kits. We agree with the reviewer that it is important to identify the mechanism(s) underlying the phenotype heterogeneity of Il33tg mice. However, this question is beyond the scope of the current study. 

 At the recommendation of the reviewer, we have added scatterplots to all bar graphs. Further, we have performed more rigorous statistical analyses of the existing data. After excluding outliers, we found that sorted ILC2s from Il33tg mice had significantly higher levels of IL-5 production (modified Fig 4). We have modified our manuscript accordingly. To clarify our message, we have also removed the data derived from total lung cell culture.

---

## [Editor Report · Decision Letter 1]

12 May 2021

Transient IL-33 upregulation in neonatal mouse lung promotes acute but not chronic type 2 immune responses induced by allergen later in life

PONE-D-20-38939R1

Dear Dr. Drake,

We’re pleased to inform you that your manuscript has been judged scientifically suitable for publication and will be formally accepted for publication once it meets all outstanding technical requirements.

Kind regards,

Svetlana P. Chapoval

Academic Editor

PLOS ONE

---

## [Editor Report · Acceptance letter]

20 May 2021

PONE-D-20-38939R1 

Transient IL-33 upregulation in neonatal mouse lung promotes acute but not chronic type 2 immune responses induced by allergen later in life 

Dear Dr. Drake:

I'm pleased to inform you that your manuscript has been deemed suitable for publication in PLOS ONE. Congratulations! Your manuscript is now with our production department. 

Kind regards, 

on behalf of

Dr. Svetlana P. Chapoval 

Academic Editor

PLOS ONE